# Prevalence, pattern and risk factors for work-related musculoskeletal disorders among Nigerian plumbers

Chidozie Emmanuel Mbada[1]*, Aanuoluwa Feyisike Abegunrin[2], Michael Ogbonnia Egwu[2], Clara Toyin Fatoye[1,3], Haruna Moda[1], Olatomiwa Falade[4], Francis Fatoye[1,5]

1 Department of Health Professions, Faculty of Health and Education, Manchester Metropolitan University, Birley Fields Campus, Manchester, United Kingdom, 2 Department of Medical Rehabilitation, College of Health Sciences, Obafemi Awolowo University, Ile-Ife, Nigeria, 3 Health and Social Care at the University Campus Oldham (UCO), Oldham, United Kingdom, 4 Royal Oldham Hospital, Oldham, Greater Manchester, United Kingdom, 5 REACH: Research, Evaluation and Analysis in Care and Health at Manchester Metropolitan University, Manchester, United Kingdom

* c.mbada@mmu.ac.uk

**Data Availability Statement:** All relevant data are within the paper and Supporting Information files

**Funding:** The author(s) received no specific funding for this work.

## Abstract

Plumbing work is more manually driven in low-and-middle income countries (LMICs), and the prevalence of work-related musculoskeletal disorders (WMSDs) among workers who engage in plumbing in LMICs may be worse than earlier reports from developed countries. This study aimed to assess the prevalence, pattern and risk factors for work-related musculoskeletal disorders (WMSDs) among Nigerian plumbers. A total of 130 consenting plumber participated in this cross-sectional study. The Nordic Musculoskeletal questionnaire and the Job Factor Questionnaire were used to assess information on prevalence and pattern of WMSDs; and perceptions regarding work-related risks factors for WMSDs. Descriptive (mean, frequency, range, percentage and standard deviation) and inferential (Chi-square) statistics were used to analyze data. Alpha level was set at p<0.05. The mean age of the respondents is 36.56 ± 10.418 years. The mean years of experience and working hours per day are 14.15 ± 9.161 years and 8.28 ± 2.512 hours. Job characteristics were mostly installation of pipes and fixtures (99.2%), equipment and fixtures prior to installation (96.9%), and testing of plumbing system for leaks (88.5%). 12-month and 7-day prevalence of WMSDS were 84.6% and 50.8%. Low-back (63.8%), neck (55.4%) and knee (50%) were the most affected body sites. Having WMSDs limits normal activities involving the low-back (32.3%), knee (25.4%) and neck (23.8%). There was significant association between 12-month prevalence of WMSDs and use of saws and pipe cutters ($\chi^2$ = 4.483; p = 0.034), while sites of affectation had significant association with 12-month and 7-day prevalence of WMSDs (p<0.05) respectively. Nigerian plumbers have a high prevalence of WMSDs affecting most commonly the low back, neck and knee. Plumbing job factors pose mild to moderate risk to developing WMSDs, and use of saws and pipe cutters significantly influence WMSDs.

**Competing interests:** The authors have declared that no competing interests exist.

## Introduction

Musculoskeletal disorders (MSDs) are generally known to cause severe long-term pain and physical disability among hundreds of millions of people around the world [1]. Globally, MSDs have shown to be the fourth greatest burden on health [2]. Storheim et al. [3] reported that a major proportion of the entire years lived with disability (YLDs) is caused by MSDs globally, second only to mental and behavioural problems [4]. The same study found MSDs as the second greatest cause of disability based on YLDs, considering a 45% increase in disability related to MSDs within two decades [4].

The risk factors of MSDs are classified into individual factors, physical factors, and psychosocial factors [5–7]. The physical risk factors of MSDS according to Ng et al. [8] include but are not limited to lifting heavy loads, awkward postures and repetitive actions while the psychosocial risk factors are work satisfaction, stress and social support [9]. Accordingly, the World Health Organization (WHO) identified several risk factors of MSDs with work-related issues greatly contributing to the conditions [10, 11]. Work-related musculoskeletal disorders (WMSDs) are reported to be a major health problem globally [12, 13]. Specifically, WMSDs is associated with cost of treatment [14], absenteeism [13], presenteeism [15], job transfer, work restriction [16, 17], loss of productivity [18], poor quality of life (QOL), and disability [16, 19]. Literature is replete on the effects of WMSDs on individuals, society and economy [20, 21].

Work-related musculoskeletal disorders cut across different occupational groups [22, 23], and their prevalence and impacts vary across the groups [23, 24]. Occupation groups whose work demands require high level of physical exertion, repetitive tasks, awkward postures and heavy lifting are said to have higher rates of MSDs [25, 26]. Less developed countries that are yet to attain the noteworthy automation and mechanization of works, are particularly prone to higher burden of WMSDs [27]. Studies have reported high rates of WMSDs among occupational groups that depend on manual approaches [28, 29].

Work-related musculoskeletal disorders are commonly faced by workers in the construction industry as a result of exposure to daily work hazards [30]. These construction workers include masons, carpenters, electricians, plumbers, amongst others [30]. While there are few studies on WMSDs among these construction workers, fewer still exist among plumbers. Plumbers are responsible for installing piping or plumbing systems [31]. The Center for Disease Control and Prevention of the United States of America (USA) reported a 40% prevalence of WMSDs among workers who engage in plumbing/pipe fitting [32]. The CDC report indicated that plumbers suffer more from WMSDs affecting the low back region (69.4%), followed by the knees (51.9%), and the least being the hips/thigh region (16.7%) [32]. Plumbing work is less manually driven in the USA, thus the extrapolation of the CDC report to less developed context is questionable. Anecdotally, plumbing in the Nigerian context is based on manual labour involving physical work done by the people, in contrast to the use of machines. While plumbers in Nigeria are relatively scarce compared with others in the construction industry, they are typically involved in repair services or in new building constructions, and their tasks involve installing or repairing pipes and fittings in corporate and residential houses. Therefore, this study aimed to assess the prevalence, pattern and risk factors for WMSDs among plumbers.

## Materials and methods

Workers who engage in plumbing jobs registered under the Association of Professional Plumbers of Nigeria (APPN) (Ile-Ife, Osun state branch) were recruited into this cross-sectional study. Eligible respondents must have been involved in active plumbing work for no less than one year. However, individuals who have sustained notable accidents or debilitating

injuries, as well as those with physical disability, psychiatric and known psychological disorders were excluded. Consenting respondents were recruited at the venue of APPN monthly meeting at the city of Ile-Ife, Osun State Nigeria.

Sample size for this study was determined using the Cochran's sample formula [33] as shown below:

$$n_0 = \frac{Z^2 \text{ x p x q}}{(e)^2}$$

Where $n_0$ signifies the sample size; z-value is found in a Z table (1.96 for a confidence level of 95%); p is the (estimated) proportion of the population which has the attribute in question (0.4); q is 1-p (1–0.4 = 0.6) and e is the desired level of precision (i.e., the margin of error = 0.05).

$$n_0 = \frac{(1.96)^2 \text{ x } 0.4 \text{ x } (1-0.4)}{(0.05)^2}$$
$$= 369$$

Using the Finite Population Correction for Proportions, the sample size ($n_0$) can be adjusted using; where N (the total number of registered plumbers in the association) = 200

$$n = \frac{n_0}{1 + \frac{(n_0 - 1)}{N}}$$
$$= \frac{369}{2.84}$$
$$= 130 \text{ plumbers.}$$

In line with the calculated sample size, a total of 130 respondents participated in this study.

## Instrument

The Nordic Musculoskeletal Questionnaire (NMQ) was used in this study. The NMQ is a standardized questionnaire that was developed by Kuorinka et al. [34] to assess regional and general MSD. This questionnaire allows for comparison of musculoskeletal disorders affecting nine body areas. It contains a body map to indicate nine symptom sites: neck, shoulders, upper back, elbows, low back, wrist or hands, hips or thighs, knees and ankles or feet. The questionnaire captures data on the incidence of musculoskeletal symptoms in the last 7 days and 12 months. It also obtains information on whether the musculoskeletal symptoms have prevented the respondent from carrying out normal daily activities, and requested for medical help in the past 12 months. The test-retest reliability of NMQ is about 0.8. Its sensitivity ranges between 66 and 92% and its specificity is between 71 and 88% [34, 35].

Job Factor Questionnaire (JFQ) was used to assess the perceptions of workers regarding fifteen work-related job factors and conditions, and their contribution to the development of WMSDs. The reliability of the JFQ has been evaluated by assessing the temporal stability in a test-retest design [36]. There is a moderate to good agreement for the job factor items and the stability of the instrument was evaluated to be 0.8 [37].

Ethical approval for the study was obtained from Health Research and Ethics Committee of the Institute of Public Health (HREC number: IPHOAU/12/1650), Obafemi Awolowo University, Ile-Ife, Nigeria. Administrative permission to conduct study was obtained from the APPN officials. The respondents were informed about the study objectives, and their written (signed)

informed consent was obtained. The Yoruba translated version of the informed consent was made available for those who had preference for or were literate in Yoruba language only (Yoruba is the local language spoken in the setting where this study was conducted).

### Data analysis

Data were summarized using the descriptive statistics of mean, frequency, range, percentage and standard deviation. Chi-square analysis was used to determine the association of prevalence of self-reported WMSD and each of age, sites of affectation, occupational characteristics, and job risk factors. Data analysis was carried out using SPSS IBM version 26 software. The alpha level was set at $p < 0.05$.

### Computation

The JFQ consists of fifteen work-related job factors which describe things at work that could contribute to job-related pain and injury. The respondents indicated the factors that represented a problem on a scale of 0–10 (0 –no problem, 10- major problem). The questionnaire items were also evaluated with responses collapsed into three categories; 0–1 –no problem, 2–7 —minor to moderate problem, and 8–10 –major problem. Individual item scores were then summed, and finally transformed to a scale of 1 to 3. 1 indicating no/low risk, 2 indicating mild/moderate risk, and 3 indicating high/severe risk. No/low risk ranged from 0 to 15, mild/moderate 16–105 while high/severe risk ranged from 106–150.

### Results

The socio-demographic and work characteristics of the respondents are presented in Table 1. All the respondents were male, and preponderantly within the age group 31–40 years (43.1%), married (68.5%) and had secondary education (73.8%). The mean age of the respondents is 36.56 ± 10.418 years. A greater percentage of the respondents (79.2%) had a work experience of 1–20 years while 50% worked for 5 to 8 hours daily. The mean years of experience and working hours per day are 14.15 ± 9.161 years and 8.28 ± 2.512 hours respectively (Table 1).

In Table 2, job factors that may predispose to the risk of WMSDs are described. Of the job characteristics, installation of pipes and fixtures (99.2%), installation of support for pipes, equipment and fixtures prior to installation (96.9%), and testing of plumbing system for leaks and other problems (88.5%) were often done while the least done was installation of heating and air conditioning systems (56.2%) as presented in Table 2.

Fig 1, show the level of risks of developing WMSDs based on the job factors. 84.6% of the respondents reported mild/moderate risk from these factors, 10.8% had no or low risk, while 4.6% reported severe or high level of risk.

Table 3 shows the percentage distribution of the prevalence of WMSDs per body sites affected based on 12- months, and 7-day recalls, as well as the rates of limitation in normal activities caused by 12-month prevalence of WMSDs. There was an 84.6% 12-month prevalence of WMSDs. Based on body sites, the most affected parts were low back (63.8%), neck (55.4%), knee (50.0%), upper back (49.2%) and shoulder (37.7%). For the 7-day prevalence of WMSDs, the total rate was 50.8%. Based on body sites, the low back (26.2%), knee (18.5% knees) and hand/wrist (14.6%) were the most affected. Having WSMDs in the course of 12 months prevented normal activities involving the lower back (25.4%), neck (23.8%), upper back (21.5%), shoulders (15.4%) and wrists/hands (14.6%).

Table 4 shows the association between 12-months prevalence of WMSDs and each of age, sites of affectation, job characteristics and level of risks for WMSDs. There was no significant association between 12-months prevalence of WMSDs and age ($\chi^2 = 2.025$; p = 0.567) as well

**Table 1. Frequency distribution of respondents' socio-demographic characteristics, years of experience and working hours per day (N = 130).**

| Variable | n (%) | Minimum | Maximum | Mean ± SD |
|---|---|---|---|---|
| Age | | | | |
| 21–30 years | 40 (30.8) | 21 | 69 | 36.56 + 10.418 |
| 31–40 years | 56 (43.1) | | | |
| 41–50 years | 17 (13.1) | | | |
| >50 years | 17 (13.1) | | | |
| Marital status | | | | |
| Single | 38 (29.2) | | | |
| Married | 89 (68.5) | | | |
| Divorced | 2 (1.5) | | | |
| Widower | 1 (0.8) | | | |
| Highest Education | | | | |
| Primary | 12 (9.2) | | | |
| Secondary | 96 (73.8) | | | |
| Tertiary | 22 (16.9) | | | |
| Years of Experience | | | | |
| 1–20 years | 103 (79.2) | 2 | 46 | 14.15 ± 9.161 |
| 21–40 years | 25 (19.2) | | | |
| >40 years | 2 (1.5) | | | |
| Working hours/day | | | | |
| 1–4 hours | 9 (6.9) | 2 | 14 | 8.28 ± 2.512 |
| 5–8 hours | 65 (50.0) | | | |
| >8 hours | 56 (43.1) | | | |

as the level of risk ($\chi^2$ = 3.021; p = 0.221), except for sites of affectation (p<0.05). Also, there was significant association between having 12-months prevalence of WMSDs and use saws and pipe cutters ($\chi^2$ = 4.483; p = 0.034).

From Table 5, there was no significant association between seven-days prevalence of WMSDS and each of age ($\chi^2$ = 3.083; p = 0.379), job characteristics done often (p > 0.05), and level of risk ($\chi^2$ = 3.068; p = 0.216), except for sites of affectation (p<0.05).

## Discussion

Work-related musculoskeletal disorders account for significant work hazards in various industries, hence placing a negative socioeconomic impact on individuals, institutions, and the society [29]. Plumbing has been identified as an occupation with high-risk activities that result in WMSDs [31, 38]. Plumbing works involve tasks like hole drilling, installing hangers for pipes, lifting materials overhead and in ground-levels and manually operating tools. These tasks expose plumbers to vibrations from tools and keep them in sustained non-neutral positions for a long time [31]. Plumbers engage in physically demanding activities due to the nature of their work that involves being in awkward postures, and repetition of strenuous tasks [30]. Plumbing work characteristics and demands vary from one context to another. These activities particularly, in less developed contexts where there is limited technology to drive the work, predispose plumbers to the risk of WMSDs. Common tasks among plumbers in most developing contexts involve installation of pipes and fixtures such as sinks and toilets, and installation of support for pipes, equipment and fixtures prior to installation.

Generally, there is still a paucity of literature on the occupational health and safety of plumbing workers. In this study, the prevalence, pattern and risk factors for work-related

**Table 2. Frequency distribution of job characteristics and those often done by the respondents (N = 130).**

| Variable | No | Yes |
|---|---|---|
| | n (%) | n (%) |
| Job Characteristics | | |
| Interprets blueprints and building specifications to map layout for pipes, drainage | 13 (10) | 117 (90) |
| Installs pipes and fixtures, such as sinks and toilets | 0 (0.0) | 130 (100) |
| Installs support for pipes, equipment and fixtures prior to installation | 0 (0.0) | 130 (100) |
| Assembles fittings and valves for installation | 5 (3.8) | 125 (96.2) |
| Modifies length of pipes, fixtures, and other plumbing materials as needed | 1 (0.8) | 129 (99.2) |
| Uses saws and pipe cutters as necessary | 3 (2.3) | 127 (97.7) |
| Installs heating and air conditioning systems including water heaters | 34 (26.2) | 96 (73.8) |
| Tests plumbing systems for leaks and other problems | 4 (3.1) | 126 (96.9) |
| Chooses plumbing materials based on budget, location and intended uses | 8 (6.2) | 122 (93.8) |
| Performs inspections and oversees other workers, such as apprentice | 17 (13.1) | 113 (86.9) |
| Job Characteristics often done | | |
| Interprets blueprints and building specifications to map layout for pipes, drainage | 41 (31.5) | 89 (68.5) |
| Installs pipes and fixtures, such as sinks and toilets | 1 (0.8) | 129 (99.2) |
| Installs support for pipes, equipment and fixtures prior to installation | 4 (3.1) | 126 (96.9) |
| Assembles fittings and valves for installation | 27 (20.8) | 103 (79.2) |
| Modifies length of pipes, fixtures, and other plumbing materials as needed | 21 (16.2) | 109 (83.8) |
| Uses saws and pipe cutters as necessary | 19 (14.6) | 111 (85.4) |
| Installs heating and air conditioning systems including water heaters | 57 (43.8) | 73 (56.2) |
| Tests plumbing systems for leaks and other problems | 15 (11.5) | 115 (88.5) |
| Chooses plumbing materials based on budget, location and intended uses | 22 (16.9) | 108 (83.1) |
| Performs inspections and oversees other workers, such as apprentice | 36 (27.7) | 94 (72.3) |

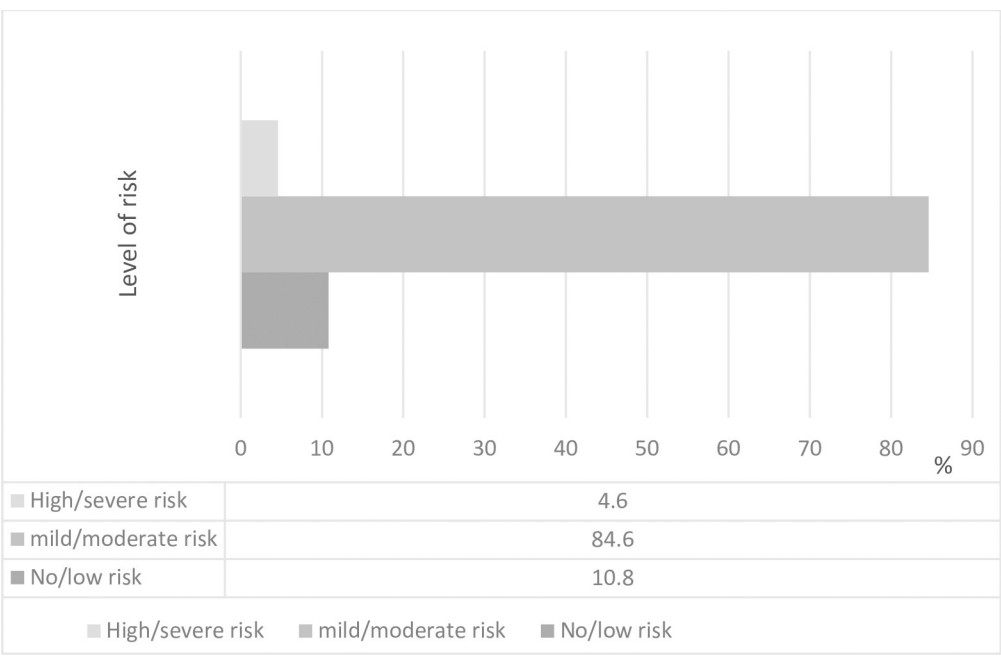

**Fig 1. Frequency distribution of the level of risk caused by the job factors among the respondents (N = 130).**

**Table 3. Percentage distribution of 12-month prevalence, 7-day prevalence and limitation in normal activities caused by 12-month prevalence of WMSDs at the regions of the body in the last twelve months (N = 130).**

| Body site | 12-month prevalence of WMSDs | 7-day prevalence of WMSDs | Limitation in normal activities by (a) |
|---|---|---|---|
| | n(%) | n(%) | n(%) |
| All body sites | | | |
| No | 20 (15.4) | 64(49.2) | |
| Yes | 110 (84.6) | 60(50.8) | |
| Neck | | | |
| No | 58 (44.6) | 114(87.7) | 99(76.2) |
| Yes | 72 (55.4) | 16(12.3) | 31(23.8) |
| Shoulders | | | |
| No | 81 (62.3) | 114(87.7) | 110(84.6) |
| Yes | 49 (37.7) | 16(12.3) | 20(15.4) |
| Upper Back | | | |
| No | 66 (50.8) | 114(87.7) | 102(78.5) |
| Yes | 64 (49.2) | 16(12.3) | 28(21.5) |
| Elbows | | | |
| No | 98 (75.4) | 116(89.2) | 116(89.2) |
| Yes | 32(24.6) | 14(10.8) | 14(10.8) |
| Wrists/Hands | | | |
| No | 86(66.2) | 111(85.4) | 111(85.4) |
| Yes | 44(33.8) | 19(14.6) | 19(14.6) |
| Lower Back | | | |
| No | 47(36.2) | 96(73.8) | 88(67.7) |
| Yes | 83(63.8) | 34(26.2) | 42(32.3) |
| Hips/Thighs | | | |
| No | 92(70.8) | 116(89.2) | 111(85.4) |
| Yes | 38(29.2) | 14(10.8) | 19(14.6) |
| Knees | | | |
| No | 65(50.0) | 106(81.5) | 97(74.6) |
| Yes | 65(50.0) | 24(18.5) | 33(25.4) |
| Ankles/Feet | | | |
| No | 97(74.6) | 112(93.8) | 112(86.2) |
| Yes | 33(25.4) | 8(6.2) | 18(13.8) |

**Table 4. Chi-square test of association between twelve-months prevalence of work-related musculoskeletal disorders and each of age, sites of affectation, job characteristics and risk factors (N = 130).**

| Variable | 12-months prevalence (general) | | | |
|---|---|---|---|---|
| | No | Yes | $\chi^2$ | p-value |
| | n (%) | n (%) | | |
| Age (years) | | | | |
| 21–30 | 8(20.0) | 32 (80.0) | 2.025 | 0.567 |
| 31–40 | 9 (16.1) | 47 (83.9) | | |
| 41–50 | 2 (11.8) | 15 (88.2) | | |
| >50 | 1 (5.9) | 16 (94.1) | | |
| Sites of Affectation | | | | |
| Neck | 0 (0) | 72 (100) | 29.342 | 0.000 |
| Shoulders | 0 (0) | 49 (100) | 4.299 | 0.000 |

*(Continued)*

**Table 4.** (Continued)

| Variable | 12-months prevalence (general) | | | |
|---|---|---|---|---|
| | No | Yes | $\chi^2$ | p-value |
| | n (%) | n (%) | | |
| Upper back | 0 (0) | 64 (100) | 22.920 | 0.000 |
| Elbows | 0 (0) | 32 (100) | 7.718 | 0.005 |
| Wrists/hands | 0 (0) | 44 (100) | 12.093 | 0.001 |
| Lower back | 0 (0) | 83 (100) | 41.741 | 0.000 |
| Hips/thighs | 0 (0) | 38 (100) | 9.763 | 0.002 |
| Knees | 0 (0) | 65 (100) | 23.636 | 0.000 |
| Ankles/feet | 0 (0) | 33 (100) | 8.041 | 0.005 |
| Job characteristics done often | | | | |
| Ibbs | 13 (14.6) | 76 (85.4) | 0.131 | 0.717 |
| Ipf | 20 (15.5) | 109 (84.5) | 0.183 | 0.669 |
| Isp | 19 (15.1) | 107 (84.9) | 0.293 | 0.588 |
| Afvi | 15 (14.6) | 88 (85.4) | 0.257 | 0.612 |
| Mlpf | 14 (12.8) | 95 (87.2) | 3.346 | 0.067 |
| Uspc | 14 (12.6) | 97 (87.4) | 4.483 | 0.034 |
| Ihac | 12 (16.4) | 61 (83.6) | 0.142 | 0.706 |
| Tps | 18 (15.7) | 97 (84.3) | 0.055 | 0.815 |
| Cpm | 15 (13.9) | 93 (86.1) | 1.097 | 0.295 |
| Pio | 11 (11.7) | 83 (88.3) | 3.536 | 0.060 |
| Level of Risk | | | | |
| No/low risk | 4 (28.6) | 10 (71.4) | 3.021 | 0.221 |
| Mild/moderate risk | 16 (14.5) | 94 (85.5) | | |
| High/severe risk | 0 (0) | 6 (100) | | |

**Key:**

Ibbs—Interprets blueprints and building specifications to map layout for pipes, drainage

Ipf—Installs pipes and fixtures, such as sinks and toilets

Isp—Installs support for pipes, equipment and fixtures prior to installation

Afvi—Assembles fittings and valves for installation

Mlpf—Modifies length of pipes, fixtures, and other plumbing materials as needed

Uspc—Uses saws and pipe cutters as necessary

Ihac—Installs heating and air conditioning systems including water heaters

Tps—Tests plumbing systems for leaks and other problems

Cpm—Chooses plumbing materials based on budget, location and intended uses

Pio—Performs inspections and oversees other workers, such as apprentice

Isp—Installs support for pipes, equipment and fixtures prior to installation

Afvi—Assembles fittings and valves for installation

Mlpf—Modifies length of pipes, fixtures, and other plumbing materials as needed

Uspc—Uses saws and pipe cutters as necessary

Ihac—Installs heating and air conditioning systems including water heaters

Tps—Tests plumbing systems for leaks and other problems

Cpm—Chooses plumbing materials based on budget, location and intended uses

Pio—Performs inspections and oversees other workers, such as apprentice

musculoskeletal disorders (WMSDs) among Nigerian plumbers were assessed. In this study, all respondents were male, and preponderantly within the age group 31–40 years, who were married and had up to secondary education. This result shows that plumbing is a male-

**Table 5. Chi-square test of association between seven-days prevalence of work-related musculoskeletal disorders and each of age, sites of affectation, job characteristics and risk factors (N = 130).**

| Variable | 7-days prevalence (general) | | | |
|---|---|---|---|---|
| | No | Yes | $\chi^2$ | p-value |
| | n (%) | n (%) | | |
| Age (years) | | | | |
| 21–30 | 21(52.5) | 19 (47.5) | 3.083 | 0.379 |
| 31–40 | 29 (51.8) | 27 (48.2) | | |
| 41–50 | 9 (52.9) | 8 (47.1) | | |
| >50 | 5 (29.4) | 12 (70.6) | | |
| Sites of Affectation | | | | |
| Neck | 0 (0.0) | 16 (100) | 17.693 | 0.000 |
| Shoulders | 0 (0.0) | 16 (100) | 17.693 | 0.000 |
| Upper back | 0 (0.0) | 16 (100) | 17.693 | 0.000 |
| Elbows | 0 (0.0) | 14 (100) | 15.214 | 0.000 |
| Wrists/hands | 0 (0.0) | 19 (100) | 21.578 | 0.000 |
| Lower back | 0 (0.0) | 34 (100) | 44.646 | 0.000 |
| Hips/thighs | 0 (0.0) | 14 (100) | 15.214 | 0.000 |
| Knees | 0 (0.0) | 24 (100) | 28.542 | 0.000 |
| Ankles/feet | 0 (0.0) | 8 (100) | 8.266 | 0.004 |
| Job characteristics done often | | | | |
| Ibbs | 41 (46.1) | 48 (53.9) | 1.130 | 0.288 |
| Ipf | 64 (49.6) | 65 (50.4) | 0.977 | 0.323 |
| Isp | 63 (50.0) | 63 (50.0) | 0.969 | 0.325 |
| Afvi | 52 (50.5) | 51 (49.5) | 0.312 | 0.576 |
| Mlpf | 53 (48.6) | 56 (51.4) | 0.099 | 0.753 |
| Uspc | 53 (47.7) | 58 (52.3) | 0.668 | 0.414 |
| Ihac | 40 (54.8) | 33 (45.2) | 2.062 | 0.151 |
| Tps | 58 (50.4) | 57 (49.6) | 0.578 | 0.447 |
| Cpm | 50 (46.3) | 58 (53.7) | 2.199 | 0.138 |
| Pio | 43 (45.7) | 51 (54.3) | 1.650 | 0.199 |
| Level of Risk | | | | |
| No/low risk | 6 (42.9) | 8 (57.1) | 3.068 | 0.216 |
| Mild/moderate risk | 57 (51.8) | 53 (48.2) | | |
| High/severe risk | 1 (16.7) | 5 (83.3) | | |

**Key:**

Ibbs—Interprets blueprints and building specifications to map layout for pipes, drainage

Ipf—Installs pipes and fixtures, such as sinks and toilets

Isp—Installs support for pipes, equipment and fixtures prior to installation

Afvi—Assembles fittings and valves for installation

Mlpf—Modifies length of pipes, fixtures, and other plumbing materials as needed

Uspc—Uses saws and pipe cutters as necessary

Ihac—Installs heating and air conditioning systems including water heaters

Tps—Tests plumbing systems for leaks and other problems

Cpm—Chooses plumbing materials based on budget, location and intended uses

Pio—Performs inspections and oversees other workers, such as apprentice

dominated occupation in Nigeria, which is in line with observations elsewhere [21]. The plumbers in this study had varied number of years of experience, up to 20 years, with a daily work demand of up to 5–8 hours. These plumbers most often engage in installation of pipes and fixtures, installation of support for pipes, equipment and fixtures prior to installation, and testing of plumbing system for leaks and other problems. Installation of heating and air conditioning systems were sparsely carried out among these plumbers.

The 12-months and seven-days prevalence of WMSDs among the plumbers were 84.6% and 50.8% respectively. The 12-month prevalence is lower than the rate of 98.1% reported among construction workers in Kenya [16] but higher than 76.8% and 77% rates reported by Merlino et al. [39] and Bodhare et al. [40] among apprentice construction workers in the USA and their counterpart in India. Other studies have reported much lower prevalence of WMSDs among construction workers in Saudi Arabia [41]. The variations in prevalence rates are probably due to context-specific differences in job characteristics and demands, studied population and methods.

In the present study, the most affected body site by WMSDs in the last twelve months was the lower back (63.8%). This finding is in line with previous reports among construction workers [42–44]. Also, the twelve-months prevalence of WMSDS of the neck (55.4%) and knees (50%) were high in this study. These findings are in agreement with those of Bodhare et al. [40] which reported high prevalence of WMSDs of the lower back, followed by the neck and knees a sample of young construction workers. Another study by Alghadir & Anwer [41] reported the low back (50%) and knee (20%) as the most affected body sites. Furthermore, findings of this study on seven-days prevalence of WMSDs show that the lower back (26.2%), followed by the knees (18.5%) and wrists/hands (14.6%) were the most hit body sites. This pattern was somewhat different from the findings by Bodhare et al. [40] where the lower back, neck and knees were the most severely affected body regions in the past seven days. In this study, an association was found between sites of affectation by WMSDs among plumbers and use of saws and pipe cutters. These findings corroborate the report of Punchihewa [45] that suggested an immediate action was necessary for work elements involving cutting pipes and preparing fittings, as well as connecting pipes and fittings in order to reduce the risk of WMSDs among plumbers.

A high proportion of the plumbers in this study reported mild/moderate risk from the various job factors, 10.8% had no or low risk from the job factors, while 4.6% reported severe or high level of risk. The risk factors contributing to WMSDs among construction workers from different studies include working in the same position for long periods [39] repetitive lifting of weights, awkward postures [46], working against force or vibration and fast work pace [14]. In a study by Punchihewa [45], having to lift and carry objects, applying forces while in awkward postures and kneeling for long durations were identified as work tasks that potentially pose high risk of MSDs among plumbers. Reporting that plumbing work poses mild to moderate risk to developing MSDs among the respondents in this study, despite its high dependence on manual labour can be attributed to 'healthy worker survivor effect (HWSE)'. The concept of HSWE refers to a selection process where workers who remain in an employment or occupation are apparently or really healthier than those who leave work [47, 48]. Based on this concept, workers who are less healthy tend to accrue less exposure than workers who remain in employment by outrightly leaving their jobs early or by reducing the amount of time that they work, by switching to lower exposed or demanding [47, 48]. Thus, it is adducible that for the plumbers in this study to have practiced for up to 14.15 + 9.161 years and 8.28 + 2.512 hours per day on the average, despite the physicality of the job, they are apparently or really healthy. It necessarily might not be because plumbing work is less demanding and exerting. Massamba et al. [48] asserts that HWSE usually leads to underestimation of the effects of harmful

occupational exposures. As HSWE generally attenuates an adverse effect of exposure [47]. It is also explainable that the plumbers in this study stayed on in this occupation for economic survival. Based on the socio-demographics, the plumbers had basic education that may have qualified them for other less demanding jobs. Therefore, overtime with the job, they may have learnt techniques about injury prevention, how to avoid harmful physical load, as well as develop better coping strategies for MSDs.

The non-probability nature of this survey limits its generalizability. However, we tried to minimize this effect, by recruiting professional plumbers who were members of their registered association. Differences exist in work characteristics and demands of these plumbers which may have influence homogeneity of sample. Furthermore, like other cross-sectional study of this nature, data on the 12 months period experience of WMSDs may be prone to recall bias, as there are chances that respondents might have given imprecise answers about their WMSDs. In addition, there is a likelihood the respondents may have attributed their MSDs as WMSDs regardless of whether they were caused by work or not. It is tricky to delineate between non-work and work-related MSDs since both are similar in presentation and consequences in response to work demands, however, work may only be a contributory factor in the aetiology of MSDs among workers [49–51].

In sum, plumbers face a high risk of WMSDs at work. Therefore, it is recommended that practical, but cost-effective measures that will minimize manual labour and physicality in plumbing work be ensured as the minimum requirement for the practice in the Nigerian and other similar contexts, especially as most of the plumbers fall within the informal sector category. Also, there is need to provide basic workplace ergonomics training on postures, lifting techniques and risk factors awareness in order to reduce risk of WMSDs among plumbers.

## Conclusion

Nigerian plumbers have a high prevalence of WMSDs affecting most commonly the low back, neck and knee. Plumbing job factors pose mild to moderate risk to developing WMSDs, and use of saws and pipe cutters significantly influence WMSDs.

## Supporting information

**S1 Questionnaire. Questionnaires used in the study (Nordic musculoskeletal questionnaire & job factor questionnaire).**
(DOCX)

**S2 Questionnaire. Questionnaire on inclusivity in global research.**
(DOCX)

## Acknowledgments

The authors thank all the Association of Professional Plumbers of Nigeria (Ile-Ife, Osun state branch) and all who volunteered to participate in the study.

## Author Contributions

**Conceptualization:** Chidozie Emmanuel Mbada, Aanuoluwa Feyisike Abegunrin, Michael Ogbonnia Egwu, Clara Toyin Fatoye, Haruna Moda, Francis Fatoye.

**Data curation:** Aanuoluwa Feyisike Abegunrin.

**Formal analysis:** Chidozie Emmanuel Mbada, Aanuoluwa Feyisike Abegunrin.

**Methodology:** Chidozie Emmanuel Mbada, Aanuoluwa Feyisike Abegunrin, Haruna Moda, Olatomiwa Falade, Francis Fatoye.

**Supervision:** Chidozie Emmanuel Mbada.

**Writing – original draft:** Chidozie Emmanuel Mbada, Aanuoluwa Feyisike Abegunrin, Michael Ogbonnia Egwu, Clara Toyin Fatoye, Haruna Moda, Olatomiwa Falade, Francis Fatoye.

**Writing – review & editing:** Chidozie Emmanuel Mbada, Aanuoluwa Feyisike Abegunrin, Michael Ogbonnia Egwu, Clara Toyin Fatoye, Haruna Moda, Olatomiwa Falade, Francis Fatoye.

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
