## [Decision Letter · Decision Letter 0]

13 Sep 2022

PONE-D-22-22802Prevalence, Pattern and Risk Factors for Work-Related Musculoskeletal Disorders among Nigerian PlumbersPLOS ONE

Dear Dr. Chidozie Mbada,

Thank you for submitting your manuscript to PLOS ONE. After careful consideration, we feel that it has merit but does not fully meet PLOS ONE’s publication criteria as it currently stands. Therefore, we invite you to submit a revised version of the manuscript that addresses the points raised during the review process.

We look forward to receiving your revised manuscript.

Kind regards,

Professor Kwasi Torpey, MD PhD MPH

Academic Editor

PLOS ONE

Journal Requirements:

Reviewers' comments:

Reviewer's Responses to Questions

**Comments to the Author**

1. Is the manuscript technically sound, and do the data support the conclusions?

Reviewer #1: No

Reviewer #2: Yes

2. Has the statistical analysis been performed appropriately and rigorously? 

Reviewer #1: No

Reviewer #2: Yes

3. Have the authors made all data underlying the findings in their manuscript fully available?

Reviewer #1: Yes

Reviewer #2: Yes

4. Is the manuscript presented in an intelligible fashion and written in standard English?

Reviewer #1: Yes

Reviewer #2: Yes

5. Review Comments to the Author

Reviewer #1: The authors need to do more rigorous work with this research. It is an interesting topic but the rationale and the justification do not sound strong. Research has been done on Musculoskeletal disorders and the link to Disability. How does it tie together? Why Plumbers? why not other artisans? It doesn't present a rich literature search proving that it is a study worth committing to.

The sample size is too small and cannot be not representative of the population of interest. Are there biases? How have they handled this?

Reviewer #2: This is an interesting, well-written, well-organized and recommendable paper that examines the prevalence of musculoskeletal problems among plumbers in low- and middle-income countries.

This research primarily sought to assess the prevalence, pattern and associated risk factors for work-related musculoskeletal disorders among plumbers. In particular, authors reported key scientific results including a 12-month and 7-day prevalence of WMSDs as 84.6% and 50.8% respectively. The authors suggested the most affected body part to be in the order of; low-back>neck> knee.

The findings are original and have not been reported elsewhere.

The choice of assessment questionnaires and study design used was appropriate in addressing the aim of this research. The analysis and presentation of results are clear and addresses the research objectives.

Authors’ conclusions clearly answered set objectives. However, it will be critical to recommend some practical, but cost-effective measures, given the high WMSD prevalences recorded among these plumbers, especially when most of them fall within the informal sector category.

The publication also followed the proper reporting requirements and community standards for data accessibility was written in clear, standard English, and complied with all applicable criteria for research ethics and integrity.

However the sentence “There is a moderate to good agreement for the job factor items and the stability of the instrument was evaluated to be 0.8 [28]” was miscited, given that paper referenced actually sought to “describe the identification and characterization of a novel human mitochondrial inner membrane protein homologous to the yeast Tim50”, hence the deviation.

6. PLOS authors have the option to publish the peer review history of their article (what does this mean?). If published, this will include your full peer review and any attached files.

Reviewer #1: No

Reviewer #2: No

---

## [Author Response · Author response to Decision Letter 0]

6 Oct 2022

PONE-D-22-22802

Prevalence, Pattern and Risk Factors for Work-Related Musculoskeletal Disorders among Nigerian Plumbers

Dear Editor,

Thank you for your careful consideration and review of our manuscript. Please find below the point-by-point responses to the comments. 

Editor Comment Response

 Please find more information on the policy and a link to download a blank copy of the questionnaire here: https://journals.plos.org/plosone/s/best-practices-in-research-reporting. Please upload a completed version of your questionnaire as Supporting Information when you resubmit your manuscript. This has been done

 We note that you have indicated that data from this study are available upon request. PLOS only allows data to be available upon request if there are legal or ethical restrictions on sharing data publicly. For more information on unacceptable data access restrictions, please see http://journals.plos.org/plosone/s/data-availability#loc-unacceptable-data-access-restrictions Data will be made available. There is no legal restriction on sharing. 

 Please review your reference list to ensure that it is complete and correct. If you have cited papers that have been retracted, please include the rationale for doing so in the manuscript text, or remove these references and replace them with relevant current references. Any changes to the reference list should be mentioned in the rebuttal letter that accompanies your revised manuscript. If you need to cite a retracted article, indicate the article’s retracted status in the References list and also include a citation and full reference for the retraction notice. This has been done. Corrected references are presented in red. Replaced references are in bold format (red).

 The manuscript must describe a technically sound piece of scientific research with data that supports the conclusions. Experiments must have been conducted rigorously, with appropriate controls, replication, and sample sizes. The conclusions must be drawn appropriately based on the data presented. This has been done. 

Reviewer 1 

 The authors need to do more rigorous work with this research. It is an interesting topic but the rationale and the justification do not sound strong. Research has been done on Musculoskeletal disorders and the link to Disability. How does it tie together? Why Plumbers? why not other artisans? It doesn't present a rich literature search proving that it is a study worth committing to. This has been done (last paragraph of the introduction section – for the rationale)

 The sample size is too small and cannot be not representative of the population of interest. Are there biases? How have they handled this? Anecdotally, plumbers in Nigeria are relatively scarce. This information has been included in the manuscript.

Reviewer 2 

 This is an interesting, well-written, well-organized and recommendable paper that examines the prevalence of musculoskeletal problems among plumbers in low- and middle-income countries. 

 This research primarily sought to assess the prevalence, pattern and associated risk factors for work-related musculoskeletal disorders among plumbers. In particular, authors reported key scientific results including a 12-month and 7-day prevalence of WMSDs as 84.6% and 50.8% respectively. The authors suggested the most affected body part to be in the order of; low-back>neck> knee.

The findings are original and have not been reported elsewhere.

The choice of assessment questionnaires and study design used was appropriate in addressing the aim of this research. The analysis and presentation of results are clear and addresses the research objectives. 

 Authors’ conclusions clearly answered set objectives. However, it will be critical to recommend some practical, but cost-effective measures, given the high WMSD prevalences recorded among these plumbers, especially when most of them fall within the informal sector category.

The publication also followed the proper reporting requirements and community standards for data accessibility was written in clear, standard English, and complied with all applicable criteria for research ethics and integrity.

However the sentence “There is a moderate to good agreement for the job factor items and the stability of the instrument was evaluated to be 0.8 [28]” was miscited, given that paper referenced actually sought to “describe the identification and characterization of a novel human mitochondrial inner membrane protein homologous to the yeast Tim50”, hence the deviation. The suggested recommendation has been included in the discussion section.

The reference [28] has been confirmed. It is presently reference [37] in this revision. 

Editor 

All new inclusion are presented in bold format. References were reformatted to suit the journal style. The questionnaires and data have been uploaded as appendix.

Thank you.

Chidozie Mbada

---

## [Editor Report · Decision Letter 1]

10 Oct 2022

Prevalence, Pattern and Risk Factors for Work-Related Musculoskeletal Disorders among Nigerian Plumbers

PONE-D-22-22802R1

Dear Dr. Mbada,

We’re pleased to inform you that your manuscript has been judged scientifically suitable for publication and will be formally accepted for publication once it meets all outstanding technical requirements.

Kind regards,

Professor Kwasi Torpey, MD PhD MPH

Academic Editor

PLOS ONE
---

## [Editor Report · Acceptance letter]

17 Oct 2022

PONE-D-22-22802R1 

Prevalence, Pattern and Risk Factors for Work-Related Musculoskeletal Disorders among Nigerian Plumbers 

Dear Dr. Mbada:

I'm pleased to inform you that your manuscript has been deemed suitable for publication in PLOS ONE. Congratulations! Your manuscript is now with our production department. 

Kind regards, 

on behalf of

Professor Kwasi Torpey 

Academic Editor

PLOS ONE